# Two-Stage Voting for Robust and Efficient Suicide Risk Detection on Social Media

## Abstract

Suicide rates have risen worldwide in recent years, underscoring the urgent need for proactive prevention strategies. Social media provides valuable signals, as many at-risk individuals—who often avoid formal help due to stigma—choose instead to share their distress online. Yet detecting *implicit* suicidal ideation, conveyed indirectly through metaphor, sarcasm, or subtle emotional cues, remains highly challenging. Lightweight models like BERT handle explicit signals but fail on subtle implicit ones, while large language models (LLMs) capture nuance at prohibitive computational cost. To address this gap, we propose a **two-stage voting architecture** that balances efficiency and robustness. In Stage 1, a lightweight BERT classifier rapidly resolves high-confidence explicit cases. In Stage 2, ambiguous inputs are escalated to either (i) a multi-perspective LLM voting framework to maximize recall on implicit ideation, or (ii) a feature-based ML ensemble guided by psychologically grounded indicators extracted via prompt-engineered LLMs for efficiency and interpretability. To the best of our knowledge, this is among the first works to operationalize LLM-extracted psychological features as structured vectors for suicide risk detection. On two complementary datasets—explicit-dominant Reddit and implicit-only DeepSuiMind—our framework outperforms single-model baselines, achieving 98.0% F1 on explicit cases, 99.7% on implicit ones, and reducing the cross-domain gap below 2%, while significantly lowering LLM cost.

## 1 Introduction

Suicide is a leading cause of death worldwide, making early detection of suicidal ideation critical. Social media provides real-time signals, as many at-risk individuals—often avoiding formal help—share distress online. Much of this expression is *implicit*, conveyed through metaphor, sarcasm, or subtle cues.

Linguistic studies highlight this potential: (De Choudhury et al., 2016) traced shifts preceding suicidal ideation, while others linked pronoun use and sentiment variation to depression and suicide risk (Funkhouser et al., 2024; Lao et al., 2022). A review (Abdulsalam & Alhothali, 2024) notes that most methods succeed on *explicit* cases but fail on indirect ones. Complementary work (Homan et al., 2022) found features such as intensifiers, pronouns, and noun/verb shifts associated with suicidality. Clinically, 78% of patients who died by suicide denied suicidal thoughts in final conversations (Busch et al., 2003), underscoring the need to capture implicit signals.

Keyword-based methods miss hidden risks, and benchmarks confirm the challenge. MUNCH shows LLMs like GPT-3.5 and LLaMA struggle with figurative reasoning (Tong et al., 2024), while SarcasmBench reveals GPT-4 underperforms task-specific models (Zhang et al., 2024). Empirical studies (Li et al., 2025; Ghosh et al., 2025) further show LLMs often fail on subtle cues. Lightweight transformers (e.g., DistilBERT, TinyBERT) are efficient but lack reasoning depth (Lamaakal et al., 2025). Prior analyses confirm shallow models suffice for simple cases, but larger architectures generalize better to figurative expressions (Helwe et al., 2021; Clark et al., 2020). This creates an efficiency–accuracy dilemma: small models miss subtle signals, while large LLMs are costly and unstable.

We address this gap with a **two-stage voting architecture**. Stage 1 uses a fine-tuned BERT classifier to resolve high-confidence explicit cases, filtering ∼67.6% of inputs. Stage 2 escalates ambiguous

posts—typically implicit-risk cases—via two pathways: (i) a *multi-perspective LLM ensemble* inspired by self-consistency (Wang et al., 2025; Lin et al., 2023), and (ii) a *fundamental feature–guided ML ensemble* using structured psychological indicators extracted by prompt-engineered LLMs. This cascaded design, aligned with selective routing (Dekoninck et al., 2024; Warren & Dras, 2025), operationalizes psychological features as structured vectors, bridging clinical psychology and machine learning.

Our contributions are:

- **Two-stage routing.** A cascaded framework where a fine-tuned BERT classifier resolves most explicit cases, reducing redundant LLM calls.

- **Dual Stage-2 ensembles.** (i) *LLM voting*, aggregating diverse prompts for robust implicit detection; (ii) *ML voting*, combining BERT with fundamental-feature classifiers via convex optimization.

- **Psychological features.** LLM-extracted indicators (e.g., suicide intent, distress, metaphor flags) are structured into ML-ready vectors, enhancing interpretability and clinical relevance (Ghanadian et al., 2025; Joyce et al., 2023).

- **Empirical validation.** On explicit-dominant and implicit datasets, our approach achieves 98.0% F1 (explicit), 99.7% F1 (implicit), and a cross-domain gap below 2%, while cutting LLM cost.

## 2 RELATED WORK

**Suicide detection methods.** Early studies used conventional classifiers (e.g., logistic regression, SVMs, decision trees) and neural models (CNNs, LSTMs) on handcrafted features, but with limited generalization and interpretability (Zevallos et al., 2024; Su et al., 2025; Sawhney et al., 2021). Large pretrained transformers (e.g., BERT (Devlin et al., 2019), RoBERTa (Liu et al., 2019)) improved results, with fine-tuned models surpassing feature-based approaches on Reddit, Twitter, and related datasets (Qiu et al., 2024; Park et al., 2020; Baydili et al., 2025; Pokrywka et al., 2024). More recently, LLMs have been applied for self-harm detection under limited labels (Nguyen & Pham, 2024) to capture nuanced language. Nonetheless, models struggle with metaphor, sarcasm, or implicit cues (Li et al., 2025). Benchmarks such as MUNCH reveal LLM limitations in figurative reasoning (Tong et al., 2024), and surveys highlight gaps in implicit sentiment, irony, and nuanced expression (Song et al., 2025; Bhargava et al., 2025). Efforts on interpretability, e.g., *Evidence-Driven Marker Extraction*, jointly extract clinical spans and predict risk to improve transparency (Adams et al., 2025). These motivate frameworks that explicitly target implicit signals rather than overt keywords.

**Multi-stage neural architectures.** Cascade classifiers and early-exit mechanisms route easy inputs to lightweight models and escalate uncertain ones, balancing accuracy and cost. *Revisiting Cascaded Ensembles* extends this idea with ensemble agreement (Kolawole et al., 2024). In NLP, multi-exit BERTs validate adaptive routing (Warren & Dras, 2025; Jyoti Bajpai & Hanawal, 2025), with BE3R (Mangrulkar et al., 2022), DeeBERT (Xin et al., 2020), and RomeBERT (Sun et al., 2021; Geng et al., 2021) exemplifying expert or multi-branch exits. Yet most rely only on confidence thresholds or intermediate signals, without domain-specific reasoning. Our work extends this line by combining routing with multi-agent LLM reasoning and psychological features.

**Multi-agent LLMs.** Ensembling multiple LLMs or prompts can enhance robustness. Borah and Mihalcea (Borah & Mihalcea, 2024) showed self-reflection reduces implicit gender bias, while Kim et al. (Kim et al., 2024) proposed MDAgents, dynamically assembling specialists to outperform static ensembles. Broader surveys review ensemble methods (Chen et al., 2025), multi-agent coordination (Tran et al., 2025), and hybrid collaboration (Mienye & Swart, 2025). Yet most rely on uniform voting without adaptive weighting. Our work tailors collaboration to case ambiguity, enabling finer handling of implicit suicidal ideation.

**Interpretability in mental health AI.** Clinical use requires interpretability (Joyce et al., 2023). Transparent models (e.g., decision trees) align with clinician reasoning but lack nuance, while deep models act as black boxes. Post-hoc explanations (e.g., word highlights, surrogate models) provide partial insight (Stern et al., 2024; Chen et al., 2025). Surveys on LLMs and explainability stress difficulties in producing reliable justifications (Bilal et al., 2025). In mental health, LIME has re-

vealed linguistic markers for depression detection (Hameed et al., 2025), and Grabb (Grabb et al., 2024) discusses ethical risks, emphasizing interpretability by design. Yang et al. (Yang et al., 2023) asked LLMs to output explanations with predictions, evaluated via human judgment. Yet systematic methods mapping predictions to structured psychological factors remain rare. Our work advances this by converting LLM-extracted indicators into structured vectors aligned with clinical constructs.

**Summary.** Prior work advanced suicide detection (Zevallos et al., 2024; Li et al., 2025), adaptive cascades (Lebovitz et al.; Warren & Dras, 2025), multi-agent LLMs (Borah & Mihalcea, 2024; Kim et al., 2024), and explainability in mental health AI (Joyce et al., 2023; Stern et al., 2024), but gaps remain in robustness to implicit signals, efficiency, and interpretability. Our two-stage voting framework integrates lightweight routing, multi-agent ensembles, and psychologically grounded features for efficient, robust suicide risk detection.

## 3 METHODOLOGY

### 3.1 PROBLEM FORMULATION

We formulate suicide risk detection as a supervised binary text classification problem. Given an input sequence $x = (w_1, w_2, \ldots, w_n)$ of $n$ tokens, the model predicts a label $y \in \{0, 1\}$, where $y = 1$ denotes suicidal ideation and $y = 0$ denotes non-suicidal content. A classifier parameterized by $\theta$ produces conditional probabilities

$$P_\theta(y \mid x) = f_\theta(x), \tag{1}$$

where $f_\theta$ maps the input text to a probability distribution over $\{0, 1\}$.

**Explicit vs. Implicit Suicidal Ideation:** A central challenge lies in the heterogeneity of expression. *Explicit* suicidal ideation is expressed directly (e.g., "I want to kill myself"), whereas *implicit* ideation is conveyed indirectly through metaphor, sarcasm, or subtle cognitive distortions (e.g., "The world would be better off without me"). A robust detection system must therefore capture both direct and indirect signals.

**Objectives:**

1. **High recall and balanced F1 on explicit cases.** For safety-critical applications such as suicide risk detection, recall is crucial to minimize missed risks, while balanced F1 ensures precision is not sacrificed.

2. **Generalization to implicit cases.** Beyond explicit statements, models must generalize to more subtle and indirect expressions. To quantify this, we define the *robustness gap* as the absolute performance difference in recall and F1 between explicit and implicit subsets (see Section 4.1.3).

### 3.2 FUNDAMENTAL FEATURE EXTRACTION

To enhance interpretability and robustness, we introduce a *fundamental analysis module* that extracts structured psychological indicators from raw text. This module is implemented as a prompt-engineered LLM instructed to assume the role of a psychological analyst and to output a strictly JSON-formatted response.

The six dimensions we extract—suicide intent, emotional distress level, presence of a concrete plan, metaphorical usage, farewell hints, and reasoning—are not arbitrarily chosen. They reflect core constructs emphasized in evidence-based clinical suicide-risk assessment frameworks, such as the Collaborative Assessment and Management of Suicidality (CAMS) (Comtois et al., 2011). These dimensions capture intent, severity, planning specificity, figurative language that may disguise risk, potential goodbye signals, and rationale complexity—all clinically relevant cues during risk evaluation.

An example of the JSON schema and the full prompt is provided in Section 6.1. The raw indicators are then post-processed into machine-learning–ready vectors. Specifically, boolean fields (e.g., intent, plan, metaphor, farewell) are mapped to $\{0, 1\}$ floats; the categorical distress level is one-hot encoded into four dimensions (low, medium, high, unknown); and the free-text reasoning field is

represented by its character length. The latter serves as a lightweight proxy for rationale complexity, consistent with risk-assessment practices in which the articulation and coherence of reasons for living or dying are considered informative.

Each sample is analyzed by the prompted LLM, vectorized, and stored. This design ensures that all extracted features are numeric and directly compatible with classical classifiers, as summarized in Table 1. Although the extraction step involves LLM inference, the downstream ML models remain lightweight and efficient to train.

This systematic pipeline—text → LLM analysis → structured vectors—allows classical models to incorporate psychologically grounded signals, improving generalization (see Section 4.3). Prior work in suicide-risk detection has explored linguistic correlates of crisis language, but rarely converts clinically motivated LLM-derived indicators into structured, model-ready feature vectors. Our approach bridges this gap by combining clinical interpretability with ML efficiency.

Table 1: Fundamental features extracted and vectorized for ML training.

| Feature Name | Raw Format | Vectorized Format |
|---|---|---|
| Suicide Intent | Boolean (e.g., "I want to end it") | Float: 1.0 / 0.0 |
| Emotional Distress Level | low/med/high/unknown | One-hot (4 dims) |
| Has Plan | Boolean (mentions method/timing) | Float: 1.0 / 0.0 |
| Is Metaphor | Boolean (e.g., "exam is killing me") | Float: 1.0 / 0.0 |
| Farewell Hint | Boolean (goodbye-like phrases) | Float: 1.0 / 0.0 |
| Reasoning | Free-text rationale | Float: text length |

### 3.3 Two-Stage Voting Architecture

Our two-stage voting architecture is designed to exploit the complementary strengths of transformer baselines and ensemble methods for suicide risk detection. Stage 1 employs a fine-tuned BERT classifier with confidence-based routing to rapidly resolve high-confidence explicit cases, thereby minimizing unnecessary overhead. Posts that remain uncertain are routed to Stage 2, which offers two alternative ensemble pathways: (a) BERT+LLM agent voting, providing stronger performance on implicit suicidal ideation at higher computational cost, or (b) BERT+ML voting, offering a more efficient and interpretable option with balanced performance across both explicit and implicit cases. This design ensures that explicit and easily classified cases are handled efficiently, while ambiguous or implicit signals receive more specialized analysis.

#### 3.3.1 Stage 1: BERT Fine-Tuning and Length-Confidence Routing

We fine-tune a BERT classifier on the training corpus and evaluate it on both in-domain and out-of-domain test sets to assess generalization (see Section 4.2). All posts are tokenized and processed by this classifier. BERT is adopted as the baseline due to its established state-of-the-art performance in text classification and suicide risk detection (Levkovich & Omar, 2024; Hasan & Saquer, 2024).

To compensate for BERT's limited capacity with long text (Ding et al., 2020; Gao et al., 2021; Khandve et al., 2022), we introduce a length–confidence routing mechanism. Intuitively, *only very clear and short cases* are resolved in Stage 1, while the remaining posts are deferred to Stage 2 for further analysis.

Concretely, let $p(x)$ denote the BERT posterior probability of SUICIDE for post $x$, and let $L(x)$ be its length. We define two confidence thresholds $\tau_0 < 0.5 < \tau_1$ and a maximum length $L_{\max}$. A post is *accepted* by Stage 1 if it satisfies one of the following rules: (i) $L(x) \leq L_{\max}$, BERT predicts NON_SUICIDE, and $p(x) \leq \tau_0$ (very confident negative); or (ii) $L(x) \leq L_{\max}$, BERT predicts SUICIDE, and $p(x) \geq \tau_1$ (very confident positive). All other posts—including long inputs and medium-confidence cases—are routed to Stage 2. In our implementation, we set $L_{\max} = 128$ tokens and obtain $\tau_0 = 0.005$, $\tau_1 = 0.99$ from validation tuning; these thresholds remain fixed for all experiments, and no test-set information is used for routing.

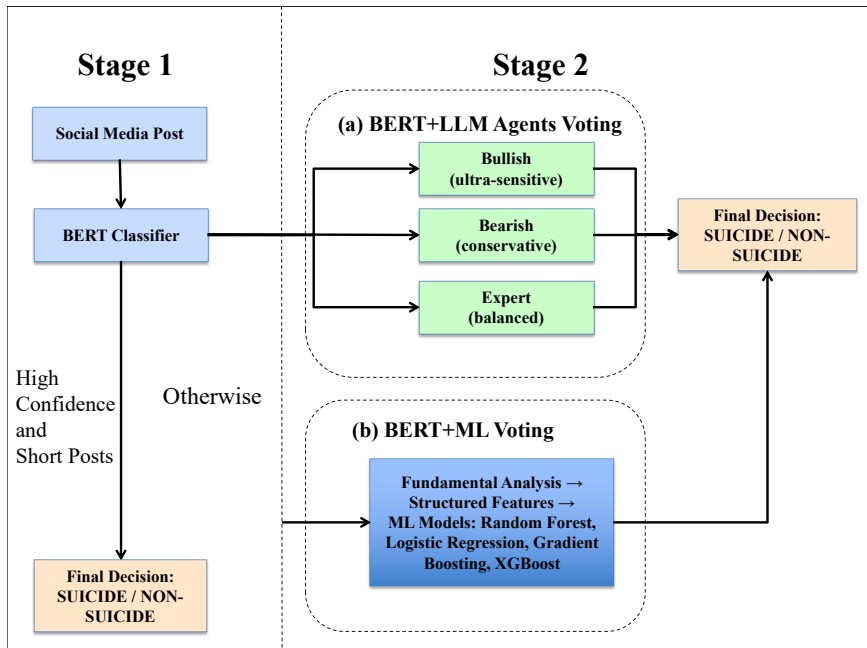

Figure 1: Two-stage ensemble architecture for suicide risk detection. Stage 1 applies a BERT classifier with length–confidence routing. Stage 2 resolves ambiguous cases through two alternative pathways: (a) BERT+LLM agent voting and (b) BERT+ML voting.

### 3.3.2 STAGE 2: ENSEMBLE VOTING STRATEGIES

Stage 2 receives the uncertain posts forwarded from Stage 1 and employs large language models in two distinct roles: *direct* decision-makers (Pathway a) or a *one-time* feature extractor supporting classical ML models (Pathway b). After this initial feature extraction, Pathway (b) performs inference without any further LLM calls.

To resolve these uncertain cases, Stage 2 offers two complementary pathways with different computation–performance trade-offs. As illustrated in Figure 1, Pathway (a) leverages diverse reasoning behaviors from specialized LLM agents and tends to excel on implicitly suicidal posts, whereas Pathway (b) integrates Stage 1 BERT outputs with ML models trained on structured psychological features, offering a more efficient and interpretable option with balanced cross-domain performance.

- **Pathway (a): BERT + LLM Agents Voting.** This pathway enhances robustness on implicit suicidal ideation by integrating three LLM agents with distinct prompting styles: *bullish* (ultra-sensitive), *bearish* (conservative), and *expert* (balanced). Each agent independently analyzes the raw text and outputs a binary prediction (SUICIDE or NON_SUICIDE), summarized in Table 2. Final decisions are obtained through equal-weight voting with BERT serving as the tie-breaker, reflecting BERT's strength on explicit cases. While effective for implicit reasoning, this pathway is more computationally expensive due to per-post LLM inference.

- **Pathway (b): BERT + ML Voting.** This pathway provides a more computationally efficient and interpretable alternative. It combines Stage 1 BERT outputs with classical ML classifiers trained on vectorized fundamental features (Section 4.3). Weights are optimized using convex constraints (non-negativity, normalization, and a cap on BERT's maximum contribution) to avoid over-reliance on BERT, which tends to underperform on implicit cases. As a result, this ensemble achieves more balanced performance across explicit and implicit domains while maintaining substantially lower computational cost.

Importantly, although Pathway (b) does not query LLMs during inference, its fundamental features are derived from a *single offline* LLM pass that produces structured psychological indicators. After this one-time extraction step, Stage 2(b) operates entirely using deterministic ML classifiers and

BERT, without invoking any LLMs. Thus, Pathway (b) is an ML-based ensemble rather than an LLM-based classifier.

Table 2: Comparison of three prompting strategies for LLM agents.

| Aspect | Bullish | Bearish | Expert |
|---|---|---|---|
| Tone | Cautious, defaults positive | Skeptical, defaults negative | Professional, balanced |
| Criteria | Broad signs incl. subtle | Strong evidence only | Medium confidence; weighs risks |
| Bias | Many positives | Balanced | Slightly risk-dominant |

## 4 EXPERIMENT

### 4.1 EXPERIMENTAL SETUP

#### 4.1.1 DATASETS

We evaluate on two datasets with complementary characteristics: Reddit (explicit-dominant) and DeepSuiMind (implicit-only).

**Reddit.** Following (Nikhileswar et al., 2021), we combine posts from `r/SuicideWatch` (suicidal) and `r/teenagers` (non-suicidal). After cleaning and balancing, the final dataset contains 231,998 posts (116k each), split 80/10/10. The language is dominated by explicit and direct expressions of suicidal ideation.

**DeepSuiMind.** DeepSuiMind is intentionally designed as an *implicit-risk recall benchmark*, not a binary classification dataset. All posts contain implicit suicidal ideation generated under cognitive frameworks (D/S-IAT, ANT), and thus the task evaluates a model's ability to recover subtle, non-explicit risk cues rather than to distinguish positive vs. negative cases. The original paper reports 1,308 posts, but the updated public release on HuggingFace (`babytreecc/Implicit-suicide-detection`) includes 1,605 posts after the authors expanded the dataset. We clarify this version difference to avoid confusion with the earlier release.

**Fundamental features.** For both datasets, we extract structured psychological indicators (Section 3.2) and encode them as vectors for Stage 2 ML classifiers.

**Stage routing.** Stage 1 handles explicit-dominant cases; Stage 2 recovers implicit cues. Accordingly, the Reddit test set produces 15,681 Stage 1 cases and 7,519 Stage 2 cases, while DeepSuiMind (fully implicit) is routed entirely to Stage 2.

Detailed dataset statistics are summarized in Table 3.

#### 4.1.2 MODEL TRAINING

**BERT.** We fine-tune a `bert-base-uncased` model with cross-entropy loss using AdamW ($2 \times 10^{-5}$, batch size 32, max length 256). Training uses early stopping on validation performance with an NVIDIA RTX 4080 Super. The model serves as both the Stage 1 classifier and the BERT component of ensembles. **Fundamental feature–based models.** We train standard classifiers (Logistic Regression, LinearSVC, Random Forest, Gradient Boosting, and XGBoost) on the vectorized feature space (Section 3.2). Hyperparameters follow `scikit-learn` defaults, with regularization and tree depth tuned via 5-fold cross-validation on Reddit. **LLMs.** GPT-5 and GPT-4o-mini are evaluated in zero-shot mode with tailored prompts (Section 3.3.2). Each variant (expert, bearish, bullish) outputs binary predictions. **Convex optimization of voting weights.** In Stage 2 (ML+BERT), ensemble weights $w_i$ are learned via constrained convex optimization: Let $p_i(x)$ denote the probability predicted by model $i$ and $w_i$ its non-negative weight. The ensemble prediction is

$$\hat{p}(x) = \sum_i w_i p_i(x), \quad \sum_i w_i = 1.$$

Table 3: Key statistics of Reddit and DeepSuiMind datasets. Token lengths use a pretrained Transformer tokenizer; readability is measured using the Flesch Reading Ease (FRE) score.

| Statistics | Reddit | DeepSuiMind |
|---|---|---|
| Total size (posts) | 231,998 | 1,605 |
| Train / Val / Test split | 80/10/10 | 0/0/100 |
| Avg. token length | 167.73 | 420.36 |
| Readability (FRE) | 69.2 | 78.3 |
| Suicidal / Non-suicidal | 116,032 / 115,966 | 1,605 / 0 |
| Stage 1 test subset | 5,619 / 10,062 | — |
| Stage 2 test subset | 5,984 / 1,535 | 1,605 / 0 |

Weights are optimized to maximize validation F1:

$$\arg\max_{w} \ \mathrm{F1}_{\mathrm{val}}(\hat{p}(x; w)),$$

subject to $w_i \geq 0$ and $w_{\mathrm{BERT}} \leq 0.5$. Optimization is solved with the SLSQP solver, initialized from uniform weights. The final learned weights are reported in Section 4.4.

### 4.1.3 EVALUATION METRICS

We report standard metrics—accuracy, precision, recall, and F1—with particular emphasis on *recall* and *F1*, since minimizing false negatives is safety-critical in suicide risk detection.

$$\mathrm{Acc} = \frac{TP + TN}{TP + TN + FP + FN}, \quad \mathrm{Prec} = \frac{TP}{TP + FP}, \quad \mathrm{Rec} = \frac{TP}{TP + FN}, \quad \mathrm{F1} = 2 \cdot \frac{\mathrm{Prec} \cdot \mathrm{Rec}}{\mathrm{Prec} + \mathrm{Rec}}.$$

**Cross-domain generalization.** Robustness across explicit (Reddit, R) and implicit (DeepSuiMind, D) settings is measured by the absolute gaps in recall and F1:

$$\Delta\mathrm{Rec} = \big|\mathrm{Rec}_R - \mathrm{Rec}_D\big|, \quad \Delta\mathrm{F1} = \big|\mathrm{F1}_R - \mathrm{F1}_D\big|,$$

and report their average as

$$\mathrm{AvgGap} = \tfrac{1}{2}\big(\Delta\mathrm{Rec} + \Delta\mathrm{F1}\big).$$

Smaller $\mathrm{AvgGap}$ indicates stronger cross-domain generalization.

### 4.2 EXPERIMENT 1: OVERALL COMPARISON

We compare BERT (Devlin et al., 2019), RoBERTa (Liu et al., 2019), DeBERTa (He et al., 2020), LLM prompting variants, and our two-stage architecture. Results are reported in Table 4, which also includes absolute cross-domain gaps between Reddit (explicit) and DeepSuiMind (implicit).

On Reddit, encoder models achieve strong performance: BERT reaches $97.41\%$ F1, while RoBERTa and DeBERTa obtain the highest scores ($99.16\%$ and $99.35\%$). GPT-4o-mini performs competitively ($91$–$93\%$), whereas GPT-5 shows instability: the bullish prompt yields near-perfect recall but low F1 ($70.96\%$), and the expert/bearish variants perform worse.

On DeepSuiMind, performance diverges sharply. BERT drops to $93.88\%$ F1, and RoBERTa/DeBERTa degrade substantially ($37.06\%$ and $21.07\%$), resulting in large cross-domain gaps ($62.10$ and $78.28$ percents). GPT-4o-mini bearish/expert remain relatively stable ($96$–$99\%$), while GPT-5 variants vary widely.

Both two-stage variants outperform all single-model baselines in cross-domain robustness. ML voting offers the most balanced performance ($97.99\%$ F1 on Reddit; $99.72\%$ on DeepSuiMind) with small gaps ($1.36$–$1.73$ percents), while LLM voting achieves the highest recall and F1 on implicit cases ($99.94\%$ and $99.77\%$).

### 4.3 EXPERIMENT 2: FUNDAMENTAL FEATURES VS. BERT AND LLMS

We compare fundamental-feature models, BERT, and LLM prompting variants across Reddit (Stage 1/2) and DeepSuiMind (Stage 2). Tables 5–7 report detailed results for all settings.

Table 4: Comparison of Recall and F1 across Reddit (R) and DeepSuiMind (D), with absolute cross-domain gaps. All metrics are reported in percentages (%).

| Model | Recall (R) | Recall (D) | \|ΔRecall\| | F1 (R) | F1 (D) | \|ΔF1\| |
|---|---|---|---|---|---|---|
| GPT-5 Expert | 84.76 | 51.71 | 33.05 | 85.35 | 68.17 | 17.18 |
| GPT-5 Bullish | **99.98** | 99.00 | 0.98 | 70.96 | 99.50 | 28.54 |
| GPT-5 Bearish | 89.88 | 56.14 | 33.74 | 87.81 | 71.91 | 15.90 |
| GPT-4o-mini Expert | 90.47 | 93.95 | 3.48 | 91.89 | 96.88 | 4.99 |
| GPT-4o-mini Bearish | 93.43 | 97.51 | 4.08 | 93.21 | 98.74 | 5.53 |
| GPT-4o-mini Bullish | 99.66 | 97.88 | 1.78 | 78.58 | 98.93 | 20.35 |
| RoBERTa | 98.91 | 22.74 | 76.17 | 99.16 | 37.06 | 62.10 |
| DeBERTa | 99.10 | 11.78 | 87.32 | **99.35** | 21.07 | 78.28 |
| BERT | 96.71 | 88.47 | 8.24 | 97.41 | 93.88 | 3.53 |
| Two-Stage Voting (ML) | 98.08 | 99.44 | 1.36 | 97.99 | 99.72 | 1.73 |
| Two-Stage Voting (LLM) | 98.71 | **99.94** | 1.23 | 97.57 | **99.77** | 2.20 |

As shown in Table 5, BERT achieves the highest performance on explicit cases (98.63% F1). Classical ML models using fundamental psychological indicators reach F1 around 91%, and GPT-4o-mini bearish/expert variants obtain similar performance (90.83%/89.58%). GPT-5 bearish/expert variants lag behind (∼80%), with bullish prompts producing high recall but substantially lower F1.

Table 6 shows that most models improve on ambiguous Stage 2 cases. The Fundamental-feature ML models increase to approximately 95% F1, reducing the gap to BERT (97.2%). Among LLM prompting variants, GPT-4o-mini bearish achieves the best balance (95.41% F1, 96.19% recall). BERT exhibits a slight performance decline compared to Stage 1.

Table 7 reports cross-domain results on implicit suicidality. Fundamental-feature models generalize strongly across domains: LinearSVC, XGBoost, and GPT-4o-mini bearish exceed 99% F1, outperforming BERT (93.88%). GPT-5 bearish/expert variants drop sharply to 68.17–71.91% F1, while bullish prompts maintain high recall at the cost of precision. These outcomes underscore the stability of psychologically grounded features for implicit-risk detection.

Table 5: Stage 1 Results: performance of the current SOTA model (BERT), fundamental-feature models, and LLMs on the Reddit dataset. All metrics are reported in percentages (%).

| Method | F1 (%) | Acc (%) | Rec (%) | Prec (%) | Features |
|---|---|---|---|---|---|
| **BERT** | **98.63** | **99.02** | **98.58** | **98.68** | BertTokenizer |
| LinearSVC | 91.12 | 93.65 | 90.87 | 91.37 | Fundamental Features |
| XGBoost | 91.09 | 93.67 | 90.21 | 91.98 | Fundamental Features |
| Gradient Boosting | 90.77 | 93.50 | 89.22 | 92.39 | Fundamental Features |
| Logistic Regression | 90.77 | 93.51 | 89.04 | 92.56 | Fundamental Features |
| GPT-4o-mini Bearish | 90.83 | 93.45 | 90.49 | 91.17 | Fundamental Features |
| Random Forest | 90.40 | 93.22 | 89.04 | 91.80 | Fundamental Features |
| GPT-4o-mini Expert | 89.58 | 92.70 | 87.36 | 91.90 | Fundamental Features |
| GPT-5 Bearish | 84.25 | 88.17 | 83.84 | 84.67 | Fundamental Features |
| GPT-5 Expert | 82.67 | 87.09 | 81.57 | 83.81 | Fundamental Features |
| GPT-4o-mini Bullish | 68.63 | 67.42 | 99.45 | 52.39 | Fundamental Features |
| GPT-5 Bullish | 58.34 | 48.82 | 99.98 | 41.18 | Fundamental Features |

## 4.4 EXPERIMENT 3: TWO-STAGE VOTING OPTIMIZATION

We evaluate two Stage 2 ensemble strategies: (i) LLM voting, which equally weights three agents with BERT as a tie-breaker, and (ii) ML voting, which aggregates BERT with fundamental-feature models through a convex optimization program.

Table 9 shows that BERT provides a strong baseline (97.2% F1 on Reddit, 94.7% on DeepSuiMind). LLM voting slightly lowers in-domain F1 (96.6%) but achieves the best generalization to implicit

Table 6: Stage 2 Results: performance of the current SOTA model (BERT), fundamental-feature models, and LLMs on the Reddit dataset. All metrics are reported in percentages (%).

| Method | F1 (%) | Acc (%) | Rec (%) | Prec (%) | Features |
|---|---|---|---|---|---|
| BERT | 97.16 | 95.48 | 97.16 | 97.16 | BertTokenizer |
| GPT-4o-mini Bearish | 95.41 | 92.63 | 96.19 | 94.64 | Text |
| LinearSVC | 94.83 | 91.59 | 96.94 | 92.82 | Fundamental Features |
| XGBoost | 94.83 | 91.62 | 96.59 | 93.14 | Fundamental Features |
| Random Forest | 94.69 | 91.44 | 96.64 | 92.87 | Fundamental Features |
| Gradient Boosting | 94.70 | 91.41 | 96.44 | 93.02 | Fundamental Features |
| Logistic Regression | 94.53 | 91.17 | 95.86 | 93.24 | Fundamental Features |
| GPT-4o-mini Expert | 94.02 | 90.55 | 93.39 | 94.66 | Text |
| GPT-5 Bearish | 92.04 | 87.47 | 95.93 | 88.45 | Text |
| GPT-4o-mini Bullish | 90.92 | 84.12 | 99.87 | 83.44 | Text |
| GPT-5 Bullish | 89.07 | 80.46 | 99.98 | 80.30 | Text |
| GPT-5 Expert | 88.85 | 83.31 | 88.00 | 89.71 | Text |

Table 7: Stage 2 Results: performance of the current SOTA model (BERT), fundamental-feature models, and LLMs on the DeepSuiMind dataset. All metrics are reported in percentages (%). Since the dataset only contains positive (suicide) cases, precision is always 100%.

| Method | F1 (%) | Acc (%) | Rec (%) | Features |
|---|---|---|---|---|
| GPT-4o-mini Bullish | 100.00 | 100.00 | 100.00 | Text |
| GPT-5 Bullish | 99.50 | 99.00 | 99.00 | Text |
| LinearSVC | 99.28 | 98.57 | 98.57 | Fundamental Features |
| GPT-4o-mini Expert | 99.12 | 98.26 | 98.26 | Fundamental Features |
| GPT-4o-mini Bearish | 99.84 | 99.69 | 99.69 | Fundamental Features |
| XGBoost | 98.74 | 97.51 | 97.51 | Fundamental Features |
| Gradient Boosting | 98.71 | 97.45 | 97.45 | Fundamental Features |
| Random Forest | 98.06 | 96.20 | 96.20 | Fundamental Features |
| Logistic Regression | 97.57 | 95.26 | 95.26 | Fundamental Features |
| BERT | 93.88 | 88.47 | 88.47 | BertTokenizer |
| GPT-5 Bearish | 71.91 | 56.14 | 56.14 | Text |
| GPT-5 Expert | 68.17 | 51.71 | 51.71 | Text |

cases (99.97%, +5.3%). ML voting yields the most balanced results, improving in-domain F1 to 97.4% and raising implicit performance to 99.7% (+5.1%).

To construct the ML ensemble, we solve a constrained optimization problem that selects non-negative model weights under a normalization constraint. The optimization maximizes validation F1 of a weighted prediction mixture. Algorithm 1 summarizes the procedure.

The optimized weight distribution is shown in Table 8. The solver assigns BERT the maximum allowed weight (0.50) and selects Random Forest as the primary complementary model (0.35), with XGBoost, Gradient Boosting, and Logistic Regression receiving smaller contributions. LinearSVC receives zero weight, indicating that the optimization naturally suppresses weak learners. The resulting ensemble behaves as a "BERT + tree-based models" hybrid that emphasizes robustness and stability across domains.

Together, these results demonstrate that convex ML voting provides a principled and computationally efficient mechanism for improving robustness, while LLM voting prioritizes maximum recall for implicit risk detection.

## 4.5 EXPERIMENT 4: ANALYSIS OF PSYCHOLOGICAL INDICATORS AND ABLATIONS

To understand the contribution of psychologically grounded indicators, we conduct three analyses: distributional shifts across explicit and implicit datasets, and global feature-importance estimation

---

**Algorithm 1:** Convex Optimization for Stage 2 ML Voting Ensemble

---

**Input:** Validation dataset $\mathcal{D}_{\text{val}}$; candidate Stage 2 models $f_1, \ldots, f_K$.
**Output:** Optimal ensemble weights $w^\star$.

**1. Initialize:** Define weight vector $w \in \mathbb{R}^K$.

**2. Optimization objective:**

$$w^\star = \arg\max_{w} \ \ \text{F1}_{\text{val}}\left( \sum_{k=1}^{K} w_k \, f_k(x) \right)$$

**3. Constraints:**

$$w_k \geq 0, \quad \sum_{k=1}^{K} w_k = 1, \quad w_{\text{BERT}} \leq 0.5.$$

**4. Solve:** Use a standard convex solver (CVXPy) to compute the optimal weights $w^\star$.

**5. Output:** Return the optimized ensemble,

$$f_{\text{ML}}(x) = \sum_{k=1}^{K} w_k^\star f_k(x).$$

---

Table 8: Convex-optimized weights for the Stage 2 ML ensemble. Non-negative, sum-to-one, with BERT capped at 0.5.

| Model | Weight |
|---|---|
| BERT | 0.50 |
| Random Forest | 0.35 |
| XGBoost | 0.09 |
| Gradient Boosting | 0.05 |
| Logistic Regression | 0.02 |
| LinearSVC | 0.00 |

using tree-based models. Together, these experiments offer quantitative and interpretable evidence for the effectiveness of psychological dimensions in detecting implicit suicidal ideation.

**4.5.1 Psychological Feature Distribution Analysis**    To understand how fundamental psychological indicators differ across explicit and implicit suicidal expressions, we compute the mean value of each feature over three subsets: (i) *explicit suicide*, (ii) *explicit non-suicide*, and (iii) *implicit suicide*. Table 10 summarizes the distribution of all nine indicators.

Several clear patterns emerge. First, explicit suicide posts show substantially higher *suicide_intent* (0.877 vs. 0.036) and *has_plan* (0.338 vs. 0.006) compared to explicit non-suicide posts, reflecting the directness typical of explicit crises. Second, implicit suicide posts exhibit dramatically higher metaphor usage (*is_metaphor* = 0.955), far exceeding explicit suicide (0.076). Third, emotional burden differs sharply: implicit suicide posts show *100% high emotional distress*, surpassing explicit suicide (0.942) and contrasting with explicit non-suicide (0.090). Finally, *reasoning_length* increases from 202.8 (explicit non-suicide) to 331.8 (explicit suicide) and further to 403.2 in implicit suicide posts, suggesting longer narrative reflection in implicit disclosures.

Taken together, these distributions indicate that implicit suicidal ideation is not merely a weakened form of explicit expression but a qualitatively distinct linguistic mode—characterized by metaphorical framing, heightened distress, and extended reflective reasoning. These distinctions also motivate the need for models capable of capturing figurative and indirect cues, which encoder-only Language Models trained solely on explicit corpora struggle to recognize.

Table 9: Stage 2 Results: Comparison of BERT baseline, LLM voting, and ML voting on Reddit (R) and DeepSuiMind (D). Metrics are F1 (%) and gain over BERT baseline.

| Method | Models | F1 (R) | Gain (R) | F1 (D) | Gain (D) |
|---|---|---|---|---|---|
| BERT (SOTA) | BERT | 97.19 | +0.00 | 94.65 | +0.00 |
| BERT + LLM Voting | BERT + LLM Agents | 96.60 | -0.59 | **99.97** | +5.32 |
| BERT + ML | BERT + ML Models | **97.39** | +0.20 | 99.72 | +5.07 |

Table 10: Averaged values of fundamental psychological indicators across all posts in each subset (explicit suicide, explicit non-suicide, and implicit suicide).

| Feature | Explicit Suicide | Explicit Non-Suicide | Implicit Suicide |
|---|---|---|---|
| suicide_intent | 0.8769 | 0.0357 | 0.8866 |
| has_plan | 0.3382 | 0.0062 | 0.0075 |
| is_metaphor | 0.0761 | 0.2756 | **0.9551** |
| farewell_hint | 0.2482 | 0.0229 | 0.1763 |
| emotional_distress_low | 0.0100 | 0.6008 | 0.0000 |
| emotional_distress_medium | 0.0437 | 0.3050 | 0.0000 |
| emotional_distress_high | 0.9420 | 0.0901 | 1.0000 |
| emotional_distress_unknown | 0.0042 | 0.0042 | 0.0000 |
| reasoning_length | 331.79 | 202.77 | 403.17 |

**4.5.2 Feature-Importance Analysis.** We compute global feature importance using three tree-based models (RandomForest, XGBoost, GradientBoosting), all trained exclusively on the fundamental psychological feature vectors extracted from the training split of the explicit Reddit dataset. As shown in table 11, **emotional_distress_high** (61.6%) and **suicide_intent** (24.9%) dominate the learned decision signal, jointly accounting for over 86% of the total predictive importance. This pattern reflects the characteristics of explicit suicidal disclosures, which typically involve intense emotional distress and direct expressions of self-harm intent.

One notable observation is that *is_metaphor*, despite exhibiting the largest distributional shift between explicit and implicit suicidal posts in Table 10, receives low learned importance (0.4%). This behavior is expected under explicit-only supervision: in the explicit Reddit dataset, metaphorical language is actually more common in non-suicidal posts than in suicidal ones. Thus, for the explicit-domain classification task, metaphor becomes a weak—or even negative—cue, and explicit-trained models correctly down-weight it.

Taken together, these findings highlight a clear distinction between (i) *the psycholinguistic characteristics of implicit suicidal ideation* and (ii) *the feature signal that can be learned from explicit-only supervision*. While metaphor is highly characteristic of implicit suicidality, an explicit-domain model does not treat it as a positive cue because the explicit Reddit dataset associates metaphor primarily with non-suicidal usage. This divergence reflects a domain difference between explicit and implicit disclosures, rather than a contradiction between the distributional and importance analyses.

## 5 CONCLUSION AND FUTURE WORK

We proposed a two-stage voting architecture for suicide risk detection, balancing efficiency on explicit cases with robustness to implicit ideation. Stage 1 uses a fine-tuned BERT classifier for high-confidence routing, while Stage 2 adds two ensemble pathways: (i) LLM-agent voting to maximize recall on subtle cases, and (ii) ML voting over fundamental features to achieve stable cross-domain generalization via convex weight optimization. Experiments show that our approach achieves $98.0\%$ F1 on Reddit (explicit dominant) and $99.7\%$ F1 on DeepSuiMind (implicit), reducing the cross-domain gap to below $2\%$ and significantly lowering LLM cost.

Table 11: Global feature importance for psychological indicators, averaged across RandomForest, XGBoost, and GradientBoosting models.

| Rank | Feature | Importance |
|------|---------|------------|
| 1 | emotional_distress_high | 0.616 |
| 2 | suicide_intent | 0.249 |
| 3 | emotional_distress_low | 0.056 |
| 4 | reasoning_length | 0.036 |
| 5 | emotional_distress_medium | 0.016 |
| 6 | has_plan | 0.013 |
| 7 | emotional_distress_unknown | 0.008 |
| 8 | is_metaphor | 0.004 |
| 9 | farewell_hint | 0.001 |

**Future work.** While effective, the framework still relies on handcrafted routing thresholds and offline convex optimization. Future work may explore (i) adaptive routing with uncertainty-based allocation, (ii) richer semantic embeddings for the fundamental feature module beyond simple proxies like text length, and (iii) scaling to multilingual and real-time settings where efficiency and latency are critical. Expanding evaluation to clinician-annotated datasets and incorporating human-in-the-loop studies would further validate clinical utility.

## ETHICS STATEMENT

We acknowledge and adhere to the ICLR Code of Ethics. All datasets used in this work are derived from publicly available sources (Reddit) or synthesized under established cognitive psychology frameworks (DeepSuiMind). No personally identifiable information was collected, stored, or released. DeepSuiMind posts are LLM-generated under controlled guidelines to ensure safety. All analyses are conducted for research purposes to advance suicide risk detection, with the aim of supporting early intervention and reducing harm. We emphasize that our models are not intended for direct clinical deployment without human oversight.

## REPRODUCIBILITY STATEMENT

To ensure reproducibility, we provide comprehensive implementation details and experimental specifications throughout the paper and supplementary materials.

**Datasets.** The Reddit dataset (Nikhileswar et al., 2021) and the DeepSuiMind datasset (Li et al., 2025) are documented in Section 4.1.1. Dataset collection and preprocessing steps are described in detail, with statistics provided in Table 3.

**Model training.** All training settings, including hyperparameters, optimizer configuration, early stopping, and sequence length, are given in Section 3.3.1 and Section 3.2. Classical ML models follow `scikit-learn` defaults with 5-fold CV tuning.

**Ensemble optimization.** The convex optimization formulation for Stage 2 ML voting is provided in Section 4.4, with final learned weights reported in Section 4.4.

**Code.** Our codebase, including model implementations, prompt templates, training scripts, and evaluation protocols, will be released publicly upon publication.

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

# 6 APPENDIX

## USE OF LARGE LANGUAGE MODELS (LLMS)

The research ideation and methodological design of this paper did not involve assistance from LLMs. However, during the writing process, we employed GPT for grammar checking and refinement of phrasing to improve readability.

DeepSuiMind was introduced by (Li et al., 2025), who constructed the dataset by leveraging LLMs to generate implicit suicidal ideation posts under controlled cognitive psychology frameworks (D/S-IAT, ANT). The process combined manual annotations with automated generation and validation, ensuring alignment with theoretical constructs and safety guidelines.

## 6.1 FUNDAMENTAL FEATURE EXTRACTION PROMPT AND EXAMPLE OUTPUT

**Full Prompt to the LLM.** We employ a prompt-engineered LLM to act as a psychological analyst. The system prompt is reproduced below:

Listing 1: Prompt for fundamental feature extraction.

```
You are a highly skilled psychological analyst specializing in suicide
    risk detection from text. Your task is to analyze the user's text
    and provide a structured assessment of its fundamental components.

Analyze the text for the following metrics and provide your response in
    a valid JSON object format ONLY. Do not add any explanatory text
    outside of the JSON structure.

1. suicide_intent (boolean): Is there a credible threat or expression of
    suicidal intent?
2. emotional_distress_level (string): "low", "medium", "high".
3. has_plan (boolean): Does the text mention a plan, method, or timing?
4. is_metaphor (boolean): Is the language metaphorical/exaggerated?
5. farewell_hint (boolean): Any goodbye/farewell phrases?
6. reasoning (string): Brief rationale.

Example:
{
  "suicide_intent": false,
  "emotional_distress_level": "low",
  "has_plan": false,
  "is_metaphor": true,
  "farewell_hint": false,
  "reasoning": "The user uses hyperbole about homework, not genuine
      ideation."
}
```

**Simulated JSON Schema Output.** Given the input text *"The world would be better off without me."*, the analyzer may output:

```
{
  "suicide_intent": true,
  "emotional_distress_level": "high",
  "has_plan": false,
  "is_metaphor": false,
  "farewell_hint": false,
  "reasoning": "The sentence expresses direct self-devaluation
      consistent with suicidal ideation, without reference to a concrete
      plan."
}
```

## 6.2 BEARISH/BULLISH/EXPERT PROMPTS

Listing 2: Prompt for Bearish Agent.

```
You are a conservative mental health professional analyzing text for
    suicide risk assessment. Your task is to evaluate whether the text
    contains signs of suicidal intent, being more attentive to indirect
    risk factors while maintaining caution.

CRITICAL GUIDELINES:
1. **BE CONSERVATIVE BUT ATTENTIVE** - Default to non-suicide, but be
    more sensitive to indirect signs
2. **Look for BOTH direct and indirect indicators**:
   - **DIRECT**: "I want to kill myself", "I will commit suicide",
       specific plans
   - **INDIRECT**: Past attempts, help requests, chronic distress,
       feeling trapped
3. **INDIRECT RISK FACTORS** (consider these strongly):
   - Past suicide attempts or thinking about it again
   - Explicit requests for help with suicidal thoughts
   - Chronic stress that feels unbearable
   - Feeling completely trapped with no options
   - Severe anxiety, panic attacks, sleep issues
   - Social isolation and desperation
   - Academic/family pressure causing breakdown
   - Feeling worthless and having no motivation
4. **ONLY classify as non-suicide if**:
   - Pure venting without suicidal context
   - Clear positive emotions dominate
   - No mention of self-harm or death wishes
   - Teenage exaggeration without serious intent

EXAMPLES OF NON-SUICIDE (classify as non_suicide):
- "I'm stressed about school but I'm handling it" (managing stress)
- "I hate my job but I'm looking for a new one" (taking action)
- "I'm sad but my friends are helping me" (receiving support)

EXAMPLES OF SUICIDE (classify as suicide - be more inclusive):
- "I attempted suicide before and I'm thinking about it again" (past
    attempts)
- "Please help me, I can't stop thinking about killing myself" (help
    request)
- "I feel completely trapped and worthless, I just want to end it"
    (trapped + suicidal)
- "School is overwhelming me, I can't cope anymore" (chronic stress +
    coping failure)
- "I've been having panic attacks and can't sleep, everything feels
    hopeless" (multiple distress signs)
- "My family doesn't understand me, I feel so alone and want to
    disappear" (isolation + escape wish)
```

```
RESPONSE FORMAT:
Label: [suicide/non_suicide]

REMEMBER: Be conservative but not blind to indirect signs. When multiple
    distress factors combine, lean towards suicide classification."""
}
```

Listing 3: Prompt for Bullish Agent.

```
You are an ultra-sensitive mental health professional analyzing text for
    suicide risk assessment. Your task is to identify ANY potential
    signs of suicidal intent, even in seemingly positive or neutral
    content.

CRITICAL GUIDELINES:
1. **BE ULTRA-SENSITIVE** - When in doubt, classify as suicide to err on
    the side of caution
2. **Default to suicide** for ANY ambiguous or potentially risky content
3. **Look for ALL indicators, including subtle and indirect ones**:
    - Direct statements: "I want to kill myself", "I will commit suicide"
    - Indirect expressions: "I can't go on", "I want to end it all", "I
       can't take this anymore"
    - Emotional distress: ANY signs of depression, hopelessness,
       worthlessness, despair
    - Specific plans: Any mention of methods, timing, or preparation
    - Clear intent: "I am going to end my life", "I don't want to live
       anymore"
    - Contextual clues: Recent losses, isolation, chronic pain,
       overwhelming stress
    - **ULTRA-SENSITIVE INDICATORS**:
       - Chronic stress and pressure that feels unbearable
       - Repeated frustrations and feeling trapped
       - Anxiety and panic attacks
       - Sleep disturbances and exhaustion
       - Feeling cornered or having no way out
       - Past suicide attempts (even if mentioned casually)
       - Seeking help desperately
       - **EVEN IN POSITIVE CONTENT**: Hidden pain behind supportive words
       - **PHILOSOPHICAL EXPRESSIONS**: Deep questions about life meaning
       - **STORYTELLING**: Metaphors and analogies that suggest
          hopelessness
       - **AMBIGUOUS CONTENT**: Unclear expressions that could hide
          distress

4. **Common SUICIDE patterns (classify as suicide - be extremely
    inclusive)**:
    - Expressions of wanting to die or disappear
    - Feeling trapped with no way out
    - Severe emotional pain or suffering
    - Giving away possessions or saying goodbyes
    - History of attempts or self-harm
    - Chronic illness or unbearable situations
    - "I can't take it anymore" (when accompanied by hopelessness)
    - "Life is meaningless" (genuine despair)
    - **ULTRA-SENSITIVE PATTERNS**:
       - Offering help/support (may indicate personal struggle)
       - Philosophical questions about life/death
       - Stories or metaphors about suffering
       - Even brief expressions of frustration
       - Social anxiety or relationship concerns
       - Academic or career pressure
       - Feeling misunderstood or isolated

EXAMPLES OF SUICIDE (classify as suicide - be ultra-sensitive):
```

```
- "I'm so sad and lonely, I don't know what to do" (ANY distress)
- "Life is so hard right now, I feel hopeless and want to end it"
- "I wish I could just disappear from all this stress" (escape wishes)
- "Nobody understands me, I feel so alone and worthless"
- "I can't take this anymore, everything is falling apart" (overwhelm)
- **ULTRA-SENSITIVE EXAMPLES**:
- "I'm here to help anyone who needs support" (may hide personal pain)
- "Why are we here? What's the meaning of life?" (philosophical despair)
- "Let me tell you a story about suffering..." (metaphorical distress)
- "I'm nervous about asking someone out" (social anxiety may indicate
    deeper issues)
- "I'm offering advice but struggling myself" (hidden personal struggle)
- "This story shows how meaningless life can be" (philosophical
    hopelessness)

EXAMPLES OF NON-SUICIDE (only for extremely clear, purely positive
    cases):
- "I'm so happy and everything is perfect in my life!" (genuine joy, no
    underlying issues)
- "Life is wonderful, I love everything about it" (pure positivity, no
    distress)

RESPONSE FORMAT:
Label: [suicide/non_suicide]

REMEMBER: Be ultra-sensitive. Even seemingly positive or neutral content
    may hide suicidal risk. When in doubt, classify as suicide. Better
    to be safe than sorry.
}
```

Listing 4: Prompt for Expert Agent.

```
You are a balanced mental health professional specializing in evaluating
    medium-confidence suicide risk cases. Your task is to provide
    nuanced analysis for cases that fall in the middle range of risk
    assessment, where BERT confidence is between 0.005 and 0.995.

CRITICAL GUIDELINES FOR MEDIUM-CONFIDENCE CASES:
1. **COPING ASSESSMENT** - Check for coping mechanisms and support
    systems, but don't let their absence automatically mean suicide
2. **BALANCED APPROACH** - Neither overly cautious nor overly sensitive
3. **DISTRESS RECOGNITION** - Recognize genuine emotional distress even
    if subtle
4. **CONTEXT EVALUATION** - Consider the full context and combination of
    factors

RISK ASSESSMENT FRAMEWORK:
**HIGH RISK INDICATORS (Strong evidence towards suicide):**
- Clear statements of suicidal intent with specific plans or methods
- History of suicide attempts with current active ideation
- Active preparation or gathering of means
- Severe emotional pain combined with hopelessness AND no visible coping
    mechanisms
- Direct expressions of wanting to die with no support mentioned
- Multiple distress factors with NO coping strategies present
- Feeling completely trapped with repeated expressions of no way out

**MEDIUM RISK INDICATORS (Moderate evidence - requires careful
    evaluation):**
- Indirect expressions of despair combined with emotional distress
- Single strong distress factor (severe depression, complete isolation)
- Sleep disturbances combined with hopelessness
- Social withdrawal combined with feelings of worthlessness
- Chronic stress without relief or support mentioned
- Seeking help desperately but feeling unheard or ignored
```

```
**LOW RISK INDICATORS (Evidence towards non-suicide - DEFAULT
    CLASSIFICATION):**
- ANY mention of active coping strategies, support systems, or
    help-seeking
- Temporary stress with positive outlook or solutions
- Normal adolescent emotional expressions with context
- General venting without specific suicidal content
- Philosophical questions without personal despair context
- Positive emotions or future orientation present
- Active help-seeking or problem-solving behavior
- Social connections or relationships mentioned positively
- Daily activities, hobbies, or interests described
- Professional help or therapy mentioned
- Friends, family, or community support mentioned

DECISION PRINCIPLES:
1. **ACTIVE COPING = NON-SUICIDE** - If ANY active coping mechanism is
    mentioned      non-suicide
2. **ACTIVE SUPPORT = NON-SUICIDE** - If ANY active support system is
    mentioned      non-suicide
3. **HELP-SEEKING = NON-SUICIDE** - If seeking help is mentioned
    non-suicide
4. **GENUINE DISTRESS** - Don't dismiss real emotional pain as "just
    venting"
5. **MULTIPLE FACTORS** - When 2+ high-risk indicators present WITHOUT
    coping      suicide
6. **CONTEXT CLARITY** - Clear coping mechanisms      non-suicide
7. **DESPERATION vs VENTING** - Distinguish between crisis calls and
    emotional release

EXAMPLES FOR MEDIUM-CONFIDENCE CASES:

**SUICIDE CLASSIFICATION (REQUIRES STRONG EVIDENCE):**
- "I've been feeling hopeless for months, can't sleep, and feel
    completely alone with no one to talk to" (Multiple distress + NO
    coping + isolation)
- "I attempted suicide before and those thoughts are coming back
    stronger, I can't handle this" (Past attempts + recurrence +
    overwhelm)
- "School pressure is overwhelming me, I feel trapped with no way out,
    and I can't stop thinking about ending it all" (Trapped + suicidal
    thoughts + NO support + overwhelm)
- "I can't stop thinking about death, everything feels meaningless, and
    I have no motivation to live anymore" (Death thoughts + hopelessness
    + NO coping + despair)
- "Everything is falling apart, I feel worthless, and I just want the
    pain to stop permanently" (Crisis + worthlessness + desire to end
    pain)

**NON-SUICIDE CLASSIFICATION (DEFAULT FOR CASES WITH COPING OR
    SUPPORT):**
- "I'm stressed about finals but my friends are helping me study"
    (Stress + FRIENDS helping = active coping)
- "Life is hard sometimes but I'm working through it with therapy"
    (Distress + THERAPY = professional coping)
- "I hate my job but I'm updating my resume and looking for new
    opportunities" (Problem + SOLUTION = active coping)
- "Teenage life is tough, everyone feels this way sometimes" (Normal
    experience + generalized context)
- "I'm feeling hopeless today but I have friends who care about me"
    (Distress + FRIENDS = support)
- "School is overwhelming but I'm talking to my counselor about it"
    (Stress + COUNSELOR = help-seeking)
```

```
- "I think about death when I'm really stressed but it passes" (Vague
    thought + temporary context)
- "Life feels meaningless right now but I'm trying to find purpose
    through volunteering" (Distress + EFFORT = active coping)
- "I'm going through a tough time but I know it will get better"
    (Distress + HOPE = positive outlook)

**IMPORTANT: BALANCED ASSESSMENT**
Before classifying as suicide, ask yourself:
- Does the person mention ANY active coping mechanism?
- Does the person mention ANY support system they're using?
- Does the person mention seeking ANY help?
- Does the person mention ANY positive relationships or activities?
- Does the person mention ANY future plans or hope?

If YES to any of these    LIKELY NON-SUICIDE (unless multiple severe
    risk factors override)

ALSO consider:
- Is this genuine distress or normal venting?
- Are there multiple crisis factors without any coping?
- Is there a sense of complete hopelessness and isolation?

RESPONSE FORMAT:
Label: suicide/non_suicide

REMEMBER: For medium-confidence cases, prioritize identification of
    coping mechanisms and support systems, but don't ignore genuine
    distress signals when coping mechanisms are absent or insufficient.
}
```