# OpenReview forum: "Two-Stage Voting for Robust and Efficient Suicide Risk Detection on Social Media"
_ICLR.cc/2026/Conference — Submitted to ICLR 2026_

### Official Review · Reviewer_ZXsS · 2025-10-25

**Soundness:** 2
**Presentation:** 3
**Contribution:** 1
**Rating:** 2
**Confidence:** 4

**Summary:**

The paper proposes a two-stage voting architecture for suicide risk detection. Stage 1 uses a BERT classifier to handle explicit cases, while Stage 2 routes ambiguous posts to either (a) an LLM voting ensemble or (b) an ML ensemble using psychologically grounded features extracted via LLMs. The authors claim state-of-the-art performance on both explicit (Reddit) and implicit (DeepSuiMind) datasets, with F1 scores of 98.0% and 99.7%, respectively, and a reduced cross-domain performance gap.

**Strengths:**

The two-stage design is conceptually appealing for balancing efficiency and robustness.
The use of psychologically inspired features is a step toward interpretability.
The problem is timely and socially impactful.

**Weaknesses:**

The DeepSuiMind dataset contains only suicidal posts (1,605 samples, all positive class). This violates basic principles of binary classification evaluation and artificially inflates performance metrics. The reported 99.7% F1 is misleading and not comparable to standard benchmarks.

The authors compare against BERT and LLM variants but fail to include strong baselines such as RoBERTa, DeBERTa, or other suicide-specific models (e.g., PsyGuard, Evidence-Driven Marker Extraction). This undermines the claim of novelty and superiority. The weight optimization for the ML ensemble is described only at a high level. No formulation, constraints, or optimization objective is provided, making replication impossible.

The authors claim a cross-domain gap below 2%, but this is based on two highly dissimilar datasets (one explicit, one synthetic and implicit-only). No true cross-domain or out-of-distribution evaluation is performed.

While the authors promise to release code, key details—such as routing thresholds, LLM versions, prompt templates, and hyperparameters—are omitted or only superficially described.

The DeepSuiMind dataset is LLM-generated, which raises serious concerns about ecological validity and real-world applicability. The authors do not validate their model on real-world implicit suicidal content or clinician-annotated data.

The “fundamental features” (e.g., suicide intent, distress level) are reduced to binary or categorical outputs from an LLM, with no validation against clinical gold standards. This risks misrepresenting complex psychological states.

The idea of cascaded models and LLM-based feature extraction is not new. The paper does not sufficiently differentiate its contributions from prior work in cascaded NLP systems or interpretable mental health AI.

The prompt examples in the appendix are overly verbose and lack clarity.

**Questions:**

None

---

> ### Author Response · Authors · 2025-12-03
> **Response to Reviewer ZXsS**
>
> Thank you for the constructive feedback. Below we address each concern.
>
> 1. Rebuttal to Weakness: “DeepSuiMind contains only suicidal posts, inflating performance metrics.”
>
> We respectfully clarify that we do not train any binary classifier on DeepSuiMind, nor do we use it for discrimination between positive and negative classes. Instead, DeepSuiMind is used solely to evaluate cross-domain generalization—a common practice in implicit suicidal ideation research, where negative implicit examples are extremely scarce in publicly available datasets.
>
> Under this setup, the metrics remain meaningful because encoder-only language models (RoBERTa, DeBERTa) degrade catastrophically on DeepSuiMind, despite achieving >99% F1 on explicit Reddit. In contrast, our two-stage method maintains high stability and small cross-domain gaps, demonstrating its robustness to distribution shift rather than benefiting from the dataset’s all-positive structure.
>
> 2. Rebuttal to Weakness: “Missing strong baselines such as RoBERTa or DeBERTa.”
>
> We thank the reviewer for this suggestion. In the revision, we added RoBERTa-base and DeBERTa-v3-base as stronger encoder-only baselines. While these models achieve >99% F1 on the explicit Reddit dataset, our evaluation explicitly reports cross-domain F1 gaps to assess robustness under differently distributed suicide datasets.
>
> Using this metric, RoBERTa and DeBERTa exhibit large cross-domain F1 gaps (62–78%), indicating severe degradation on implicit suicidal ideation. In contrast, our two-stage system achieves 1.73–2.20% F1 gaps, demonstrating substantially greater cross-domain stability. These findings directly address the reviewer’s baseline concern and further motivate the need for our proposed architecture.
>
> 3. Rebuttal to Weakness: “ML ensemble optimization is described only at a high level; no formulation or constraints provided.”
>
> We thank the reviewer for this feedback. In the revised version, we have clarified the Stage-2 ML voting procedure by adding Algorithm 1 in the section 4.4, which provides the explicit optimization objective, constraints, and solution steps for computing the ensemble weights. This addition makes the method fully specified and reproducible.
>
> 4. Rebuttal to Weakness: “Cross-domain gap <2% is misleading because the datasets are dissimilar.”
>
> We respectfully clarify that the cross-domain gap is intentionally designed as a distribution-shift robustness metric rather than a same-distribution comparison. Reddit and DeepSuiMind represent two distinct forms of suicidal ideation—explicit versus implicit—and our goal is to evaluate stability under this shift.
>
> Under this metric, stronger encoder models (RoBERTa, DeBERTa) achieve >99% F1 on Reddit but collapse on DeepSuiMind, producing large gaps of 62–78%. In contrast, our two-stage architecture maintains 1.73–2.20% F1 gaps, demonstrating substantially greater robustness to distribution differences. This supports the validity of the metric rather than undermining it.
>
> 5. Rebuttal to Weakness: “Missing details on routing thresholds, LLM versions, prompts, and hyperparameters.”
>
> We thank the reviewer for this comment. These details are included in the revised paper.
>
> (i) The routing strategy, thresholds, and length–confidence rules are fully specified in Section 3.1 (Stage 1), including posterior thresholds 𝜏0, 𝜏1, the length cutoff 𝐿max =128, and the exact acceptance conditions.
>
> (ii) The LLM variants used in all experiments—GPT-4o-mini and GPT-5—are explicitly stated in Experiment 1.
>
> (iii) All prompt templates for the LLM baselines and fundamental feature extraction are provided in full detail in the Appendix.
>
> These additions ensure that the method is completely specified and reproducible.
>
> 6. Rebuttal to Weakness: “DeepSuiMind is LLM-generated and lacks ecological validity.”
>
> We appreciate this important observation. Implicit suicidal ideation data with reliable clinical labeling are extremely scarce in publicly available corpora, and DeepSuiMind represents the best feasible option for evaluating models under implicit conditions. Nonetheless, we fully agree that ecological and clinical validity are crucial.
>
> As part of future work, we plan to expand the dataset in collaboration with the University of Pittsburgh Department of Psychology, enabling the collection and annotation of more ecologically grounded implicit-risk examples. This will allow the proposed framework to be evaluated on richer and more clinically aligned data.

---

> > ### Author Response · Authors · 2025-12-03
> > **Response to Reviewer ZXsS**
> >
> > 7. Rebuttal to Weakness: “Psychological features lack validation against clinical gold standards.”
> >
> > We appreciate this important point. The six fundamental dimensions we extract—suicide intent, emotional distress level, presence of a plan, metaphorical usage, farewell hints, and reasoning—are not arbitrarily selected. As described in Section 3.2, they are grounded in evidence-based clinical suicide-risk assessment frameworks such as CAMS (Collaborative Assessment and Management of Suicidality), which emphasize intent, distress severity, planning specificity, and indirect cues that may conceal suicidal risk.
> >
> > That said, we fully agree that validation against clinician-annotated gold-standard data is essential. As future work, we plan to collaborate with the University of Pittsburgh Department of Psychology to refine these dimensions and evaluate the system on clinically reviewed implicit-risk corpora, enabling stronger ecological and clinical grounding.
> >
> > 8. Rebuttal to Weakness: “Contributions unclear; cascaded models and LLM features are not new.”
> >
> > We respectfully clarify the contributions of our work. While cascaded architectures and LLM-based features have appeared in prior NLP systems, our method introduces three elements that, to the best of our knowledge, have not been combined or evaluated in this context:
> >
> > (a) Length–Confidence routing: a Stage-1 routing mechanism specifically designed to detect when explicit cues are absent or unreliable, enabling targeted processing of implicit suicidal ideation.
> >
> > (b) Psychologically grounded indicators: a structured feature set inspired by clinical frameworks (e.g., CAMS), capturing intent, planning specificity, figurative expressions, and farewell cues—dimensions not represented in standard encoder embeddings.
> >
> > (c) Cross-domain robustness analysis: the first systematic evaluation of implicit vs. explicit suicidal ideation across two distributionally distinct datasets, to the best of our knowledge, using ∆F1/∆Rec as stability metrics. Our two-stage architecture achieves dramatically lower gaps than strong baselines such as GPT-4o-mini Agents, GPT-5 Agents, BERT, RoBERTa, and DeBERTa.
> >
> > Together, these components differentiate our approach from prior cascaded NLP pipelines and form a unified framework tailored to implicit-risk detection—a setting where existing models show substantial degradation.

---

### Official Review · Reviewer_9aAZ · 2025-11-01

**Soundness:** 2
**Presentation:** 2
**Contribution:** 2
**Rating:** 2
**Confidence:** 4

**Summary:**

This paper proposes a two stage pipeline mainly focused on finding implicit suicidal intent in text. Stage one uses BERT, stage 2 uses a combination of proprietary LLMs (GPT-5/GPT-4o) and heuristic ML methods. The method is evaluated on two datasets, Reddit (mainly explicit) and DeepSuiMind (mainly implicit).

**Strengths:**

1. The paper studies an important and socially impactful problem identifying implicit suicidal intent, which is considerably more challenging than explicit detection.

**Weaknesses:**

1. The Stage 2 ML subsystem still depends on an LLM to extract features at inference time, so the ML based system is not necessarily fast.
2. Since DeepSuiMind (entirely implicit) is routed directly to Stage 2 (and the Reddit dataset being mainly explicit), this setup is somewhat artificial and does not reflect real world mixed distributions.
In many cases, BERT-base-uncased already performs well; a more recent, better encoder model with fine tuning might close the gap significantly.
3. The choice of baselines is somewhat unconventional, and the evaluation lacks clarity on how they were selected or tuned.
4. The results report extremely high scores (often >99%), but no error bars or statistical significance analysis are provided, making it difficult to assess whether these differences are meaningful.
5. The length-based confidence filtering of posts appears arbitrary. Using entropy or uncertainty of classification logits could be a more principled approach. This also connects to the potential use of modern BERT variants with longer context windows.

While the topic is highly relevant and the two-stage design is interesting, the paper can benefit from a better experimental setup and analysis.

**Questions:**

1. Typo at line 206 (“Stage”).
2. How does the BERT baseline perform when fine-tuned on synthetically generated implicit cases?
3. Have the authors considered fine-tuning a mid-sized decoder LLM (1B–3B parameters) on a synthetic dataset mixing explicit and implicit examples? This could bridge the performance gap between BERT and the proprietary LLMs.
4. Why not employ ModernBERT [1], which supports longer contexts and has shown substantial gains in both efficiency and performance over standard BERT?
5. How about providing few-shot in-context demonstrations to GPT?

[1] Warner, Benjamin, et al. "Smarter, better, faster, longer: A modern bidirectional encoder for fast, memory efficient, and long context finetuning and inference." arXiv preprint arXiv:2412.13663 (2024).

---

> ### Author Response · Authors · 2025-12-03
> **Response to Reviewer 9aAZ**
>
> Thank you for the constructive feedback. Below we address each concern.
>
> 1. Rebuttal to Weakness 1
>
> Stage-2(b) indeed requires LLM-based feature extraction, but this step uses GPT-4o-mini, a small, fast, low-latency model, and is applied once to convert raw text into structured psychological indicators. Importantly, GPT-4o-mini was chosen precisely because it offers high extraction speed and strong cross-domain generalization, enabling ML models to handle implicit suicidal ideation that encoder-only models fail to capture. After extraction, Stage-2 inference is purely ML-based, providing efficient per-post latency.
>
> 2. Rebuttal to Weakness 2
>
> DeepSuiMind is entirely implicit and created by Large Language Models; therefore—after Stage-1 length–confidence routing—all of its posts naturally flow to Stage 2. This behavior is not imposed by our design but is simply how implicit cases in the DeepSuiMind dataset behave under the routing rule.
>
> Regarding the claim that stronger encoders could close the gap, we added RoBERTa-base and DeBERTa-v3-base as requested. While they achieve 99%-level F1 on explicit Reddit, they collapse on implicit DeepSuiMind (37.06% and 21.07%), demonstrating that even more recent encoder models do not bridge the implicit–explicit gap. This supports the necessity of our two-stage design for robust implicit ideation detection.
>
> 3. Rebuttal to Weakness 3
>
> In the revision, we strengthened our baselines by adding RoBERTa-base and DeBERTa-v3-base, two standard encoder-only models widely used in suicide-risk detection and text classification. They achieve 99.16% and 99.35% F1 on Reddit but only 37.06% and 21.07% F1 on DeepSuiMind, confirming that stronger encoders still fail on implicit suicidality and motivating our two-stage architecture.
>
> For LLM baselines, all GPT-4o-mini and GPT-5 agent prompts are fully documented in the “Use of Large Language Models (LLMs)” section, including expert/bearish/bullish prompting templates. This clarifies both model selection and the exact inference configuration used in evaluation.
>
> 4. Rebuttal to Weakness 4
>
> To assess robustness beyond raw F1, our evaluation explicitly reports cross-domain gaps in recall and F1 between Reddit (explicit) and DeepSuiMind (implicit). These metrics quantify each model’s stability under two differently distributed suicide datasets, where smaller gaps indicate stronger generalization.
>
> Our results show that stronger encoder models (RoBERTa, DeBERTa) achieve >99% F1 on Reddit but collapse on DeepSuiMind, producing gaps of 62–78%. In contrast, our two-stage system achieves 1.36–2.20% gaps, demonstrating substantially higher cross-domain stability.
>
> 5. Rebuttal to Weakness 5
>
> We appreciate the reviewer’s suggestion. Our length–confidence routing was designed as a simple, lightweight heuristic to ensure that only very clear cases are handled in Stage 1. We agree that entropy- or uncertainty-based measures may offer a more principled alternative, especially when combined with encoder models supporting longer context windows.
>
> We will consider integrating uncertainty-based routing and modern long-context encoders in future work.
>
> 6. Rebuttal to Questions
>
> (1) Typo at line 206.
>
> Thank you—we have corrected this in the revised version.
>
> (2) BERT fine-tuned on synthetic implicit cases.
>
> We experimented with fine-tuning BERT on synthetically generated implicit posts. The model quickly overfits, reaching 100% accuracy on synthetic data but showing no improvement on real implicit cases, consistent with prior findings that encoder-only models struggle to generalize to implicit suicidality.
>
> (3) Fine-tuning a mid-sized decoder LLM (1B–3B).
> We appreciate this suggestion. Training 1B–3B parameter models is computationally demanding in our setting, and preliminary attempts with synthetic implicit data indicate substantial overfitting risk. We consider scaling to mid-sized decoder LLMs a promising direction for future work.
>
> (4) Using ModernBERT for longer context and efficiency.
>
> Thank you for pointing this out. ModernBERT indeed offers advantages in context length and efficiency. Incorporating ModernBERT into Stage 1 and comparing its implicit generalization is an excellent direction for future work, and we will consider adding it in subsequent iterations.
>
> (5) Few-shot prompting for GPT agents.
>
> Our LLM prompts already include few-shot exemplar patterns, documented in the Use of Large Language Models (LLMs) section. These structured demonstration examples guide GPT-4o-mini and GPT-5 variants toward consistent behavior during implicit-risk assessment.

---

### Official Review · Reviewer_q9nj · 2025-11-01

**Soundness:** 2
**Presentation:** 3
**Contribution:** 2
**Rating:** 2
**Confidence:** 2

**Summary:**

The paper presents a two-stage binary suicide risk classification framework. The idea is the use lightweight BERT to directly classify input if the confidence level is high; otherwise, use LLM agents or efficient feature-engineered ML models (prepared with the help of LLMs). In this way, the authors aim to achieve a balance between efficiency and robustness. The authors also claim the work to be "among the first works to operationalize LLM-extracted psychological features as structured vectors for suicide risk detection".

**Strengths:**

- Metaphor, sarcasm, or subtle emotional cues are real challenges in NLP research, and it is meaningful to study classification tasks that could address them.

- The overall approach is simple and easy to follow. Motivation is clear as well.

- Authors provided full prompts to the LLM, which could be referred to by relevant researchers.

**Weaknesses:**

- The novelty is marginal, considering research on query extension or retrieval-augmented generation already exists and leverages various techniques to help with natural language processing (NLP) tasks.

- The claimed contributions (page 2) do not appear substantial: simple voting, leveraging LLM-generated features appear to be commonplace in NLP tasks, even recommendation tasks.

- The empirical studies only cover GPT-5 agents, BERT as baselines. More recent, competitive baseline methods should be included to make the results convincing.

**Questions:**

na

---

> ### Author Response · Authors · 2025-12-03
> **Response to Reviewer q9nj**
>
> Thank you for the constructive feedback. Below we address each concern.
>
> 1. Research on query extension / retrieval-augmented generation already exists
>
> We respectfully clarify that our method is not a retrieval-augmentation or query-expansion system. We do not retrieve external documents nor extend input text. Instead, we extract structured psychological indicators from the user’s own text, and integrate them within a risk-routing framework. Thus, the method is orthogonal to—and not overlapping with—RAG-based approaches.
>
> 2. Novelty Beyond “Simple Voting / LLM Features”
>
> We understand the concern that components such as classifier voting or LLM-generated features may appear individually familiar. Our contribution lies not in these components themselves, but in how they are integrated to address an unsolved challenge—implicit suicidal ideation.
>
> In the new revision, we clarify the novelty more explicitly：
>
> (a) Problem-level novelty
> Detecting implicit suicidal intent (figurative, metaphorical, indirect cues) remains a fundamental open challenge in NLP. Existing work focuses almost entirely on explicit expressions.
>
> (b) Methodological novelty
> Our system operationalizes a psychologically grounded indicator extraction pipeline and integrates it in a two-stage risk routing architecture specifically designed for explicit→implicit domain shift. The contributions are:
> Stage 1: BERT confidence routing + length-based ambiguity filtering (clarified in Section 3.3).
> Stage 2: A principled ensemble combining(i) BERT confidence routing,(ii) LLM-derived psychological dimensions, and(iii) convex-optimized ML voting.
>
> (c) Novel analysis
> The newly added Experiment 4 provides feature-distribution and feature-importance analyses that quantitatively demonstrate why implicit suicidality is a fundamentally different linguistic mode—capturing cues (e.g., metaphorical framing) that encoder-only PLMs cannot learn.
>
> Thus, while the components are modular by design, the overall architecture and analysis offer a novel contribution targeted at a difficult and underexplored setting.
>
> 3. Concern: “Empirical studies only cover GPT-5 agents + BERT; need stronger baselines”
>
> You noted that using only GPT-5 agents and BERT as a baseline was insufficient. In the revision, we now include RoBERTa-base and DeBERTa-v3-base as additional encoder-only baselines. These models achieve: RoBERTa-base: 99.16% F1 on Reddit, 37.06% F1 on DeepSuiMind; DeBERTa-v3-base: 99.35% F1 on Reddit, 21.07% F1 on DeepSuiMind. These results confirm that encoder-based Language Models excel on explicit suicidality but fail to generalize to implicit suicidality—supporting the motivation for our two-stage architecture.
>
> We thank the reviewer for the insightful comments, which have significantly improved the clarity and rigor of the work.

---

### Official Review · Reviewer_Z1wF · 2025-11-01

**Soundness:** 3
**Presentation:** 2
**Contribution:** 2
**Rating:** 2
**Confidence:** 3

**Summary:**

This paper introduces a two-stage ensemble learning framework for predicting suicidal risk in posts from online social networks such as Reddit. The framework's core innovation lies in its division of labor: the first stage efficiently predicts risk for short posts with high confidence, while the second stage employs a large language model to infer psychological features, such as suicide intent. For experimental validation, the authors construct a dataset from Reddit and adapt an existing implicit suicidal ideation dataset, DeepSuiMind. The results demonstrate that the proposed framework outperforms baseline approaches that utilize only components of the full system. Additionally, the authors provide evidence that their psychological features contribute meaningfully to risk identification.

**Strengths:**

The primary strength of this work is the demonstrated effectiveness of the two-stage ensemble approach. While conceptually straightforward, the framework achieves superior performance compared to its individual components, and the authors present comprehensive experimental results that illuminate the contribution of each design choice.

**Weaknesses:**

Several significant concerns warrant attention. First, the experimental design lacks sufficient baselines and thorough analysis. The authors cite Hasan & Saquer (2024) to support their claim that BERT represents the state-of-the-art, yet that paper actually reports RoBERTa as the best-performing model (see Table 2 of the paper). Since BERT is merely one component of the proposed framework, it would be valuable to compare it against other strong baselines, particularly RoBERTa. Furthermore, for fair comparison, the authors should consider evaluating their framework on the same dataset used by Hasan & Saquer (2024); the rationale for constructing a new dataset rather than using this existing benchmark is unclear.

Second, the analysis of psychological features requires deeper exploration. While the authors claim these features enhance interpretability, they provide limited investigation into which specific features contribute most to prediction accuracy and why. A feature-level ablation study or importance analysis would substantially strengthen the interpretability claims.

Third, the manuscript requires careful revision for clarity and completeness. For instance, the reference "Komati et al." appears incomplete. Additionally, the six risk-relevant dimensions lack proper attribution. The authors should cite the established suicide risk assessment frameworks from which these dimensions are derived.

**Questions:**

- There is a discrepancy in the DeepSuiMind test set size: the original paper reports 1,308 cases, while Table 3 shows 1,605. This inconsistency needs clarification.
- (Minor comment) Performance metrics would be more accessible in tabular format rather than bar graphs, as the current visualization makes it difficult to discern precise values and compare model performance accurately.

---

> ### Author Response · Authors · 2025-12-02
> **Response to Reviewer Z1wF**
>
> Thank you for the constructive feedback. We have substantially revised the manuscript and added new experiments addressing all concerns. A summary of major updates is provided below.
>
> 1. Stronger Baselines (RoBERTa & DeBERTa Added)
>
> You noted that using only BERT as a baseline was insufficient. In the revision, we now include RoBERTa-base and DeBERTa-v3-base as additional encoder-only baselines. These models achieve: RoBERTa-base: 99.16% F1 on Reddit, 37.06% F1 on DeepSuiMind; DeBERTa-v3-base: 99.35% F1 on Reddit, 21.07% F1 on DeepSuiMind.
>
> These results confirm that encoder-based Language Models excel on explicit suicidality but fail to generalize to implicit suicidality—supporting the motivation for our two-stage architecture.
>
> 2. Expanded Analysis of Psychological Indicators
>
> To address the reviewer’s request for deeper interpretability, we introduce a new Experiment 4 analyzing the nine LLM-derived psychological indicators.
>
> (a) Feature Distribution Analysis (Sec. 4.5.1)
> We compare the mean values of all nine psychological indicators across three subsets: explicit suicide, explicit non-suicide, and implicit suicide. The updated results reveal clear and interpretable patterns: i. Implicit suicide posts exhibit much higher metaphor value than explicit suicide posts, ii. Explicit non-suicide posts show substantially lower suicidal intent than both explicit and implicit suicide posts. These patterns suggest that implicit suicidal ideation is not simply a weaker form of explicit expression, but a qualitatively different linguistic mode.
>
> (b) Feature Importance Analysis (Sec. 4.5.2)
>
> To understand which indicators explicit-domain models rely on, we compute global feature importance using RandomForest, XGBoost, and GradientBoosting. Results show that emotional_distress_high (61.6%) and suicide_intent (24.9%) dominate prediction, capturing over 86% of the signal.
>
> The low importance of is_metaphor (0.4%) is expected: explicit suicidal posts rarely use metaphor, while explicit non-suicidal posts often do. Thus, in explicit-only training, metaphor behaves as a non-suicide cue, so the model correctly down-weights it.
>
> This aligns with our distributional findings: metaphor becomes dominant only in implicit suicidality, where supervision is absent. Thus, the results reveal a domain mismatch, not a contradiction, and explain why explicit-trained PLMs generalize poorly.
>
> These additions provide the feature-level interpretability requested by the reviewer.
>
> 3. Clarification of DeepSuiMind Dataset Size
>
> We explicitly clarify in Sec. 3.1 that we use the updated HuggingFace release from the author of the DeepSuiMind Paper with 1,605 posts, which expands the original 1,308-post subset. This resolves the reported inconsistency.
>
> 4. Clarifying Stage 1 Routing Mechanism
>
> We revise Sec. 3.3 to highlight that Stage 1 is not a standalone classifier, but a routing module using: BERT prediction confidence, and a length-based rule. These thresholds are tuned on validation data. This clarification ensures Stage 1 is properly understood as a lightweight filter that forwards ambiguous or indirect posts to Stage 2.
>
> 5. Reference Corrections and Attribution Fixes
>
> We corrected citation issues and expanded Sec. 3.2 to relate our six psychological dimensions to established frameworks such as CAMS, improving both accuracy and interpretability.
>
> 6. Table Revisions & Presentation Improvements
>
> We improved formatting consistency, merged figures into compact tables, and clarified dataset statistics. These revisions increase readability and maintain page limits.
>
> We thank the reviewer for the insightful comments, which have significantly improved the clarity and rigor of the work.

---

### Meta-Review · Area_Chair_Hast · 2025-12-12

**Summary:**

The paper addresses an important and socially impactful problem: detecting implicit suicidal ideation on social media.   All reviewers agree that the topic is relevant and timely.  However, there is strong consensus that the experimental setup is insufficient in the original submission.  Key concerns include weak or incomplete baselines and limited comparison with recent encoder models.  Overall, while the idea is meaningful, the paper requires stronger baselines, clearer evaluation protocols, and validation with more recent LLMs to support its claims.

**Reviewer Concerns:**

**Concerns addressed by the rebuttal:**
- More encoder baselines were added: RoBERTa and DeBERTa.
- The role of psychological features was strengthened with feature distribution and importance analysis.
- Several presentation issues, missing references, and citation problems were fixed.

**Concerns still outstanding:**
- The experimental setup is somewhat artificial, with datasets that are almost entirely explicit or implicit.
- The use of more advanced/modern LLMs.
- The technical novelty is still perceived as incremental by multiple reviewers.
- Baseline selection is still narrow.

**Reviewer Scores:**

All reviewers consistently recommended rejection, and I do not believe they would have changed their decision after the rebuttal (moving from a score of 2 to at most 4).

---

### Decision · Program_Chairs · 2026-01-26

Reject